# Supracellular contractility in *Xenopus* embryo epithelia regulated by extracellular ATP and the purinergic receptor P2Y2

Sagar D. Joshi[1], Timothy R. Jackson[1], Lin Zhang[1], Carsten Stuckenholz[1] and Lance A. Davidson[1,2,*]

## ABSTRACT

Extracellular signals regulate epithelial homeostasis, cell fate and the patterning of cell behaviors during embryogenesis, wound healing, regeneration and disease progression. Previous studies in our group found that cell lysate from intentionally wounded *Xenopus laevis* embryos triggers a strong but transient contraction in neighboring epithelia, whether contiguous to the wound site or in non-wounded embryos. We previously identified extracellular ATP (eATP) as a possible candidate signal. Here we test additional candidates and find that several nucleotides, including ADP, UTP and UDP, also trigger contractility. Through a temporal and spatial screen of lysate activity, an inhibitor screen and morpholino knockdown of candidate receptors, we find that contractility is mediated by a G-protein-coupled purinergic receptor, P2Y2 (P2RY2). Activated P2RY2 triggers F-actin assembly and myosin II contractility. Knockdown of P2RY2 or overexpression of mutant G protein effectors abrogate epithelial contractility when epithelia are exposed to eATP or lysate. We demonstrate that the major contributors to epithelial contractility in lysate are the extracellular nucleotide triphosphates ATP and UTP, which are sensed by P2RY2 and transduced through G proteins to contract the epithelium.

KEY WORDS: Epithelial homeostasis, Actomyosin contractility, Wound healing, Supracellular contractility, ATP, UTP, ADP, UDP, Adenosine

## INTRODUCTION

While there is widespread belief that cell and tissue mechanical processes operating during wound healing and vertebrate development are regulated at the supracellular scale, precise molecular pathways that function at that scale have been difficult to elucidate. To identify such factors, we aimed to investigate molecular pathways mediating tissue responses to wounding, particularly how responses are distributed across a field of cells that have not been directly injured (Joshi et al., 2010). These responses offered us the opportunity to identify signaling pathways that mediate this supracellular contractility. In this paper we describe a pathway involving extracellular ATP (eATP) and a G-protein-coupled receptor (GPCR), P2Y2 (P2RY2), that controls contractility in the embryonic epithelia of the frog *Xenopus laevis*.

[1]Department of Bioengineering, University of Pittsburgh, Pittsburgh, PA 15260, USA. [2]Department of Computational and Systems Biology, University of Pittsburgh, Pittsburgh, PA 15260, USA.

*Author for correspondence (lad43@pitt.edu)

C.S., 0000-0001-8732-4435; L.A.D., 0000-0002-2956-0437

## RESULTS AND DISCUSSION

### Cell lysate drives acute contraction of *Xenopus* embryonic epithelia

Acutely applied cell lysate can drive short-term contraction of embryonic epithelia (Joshi et al., 2010). We first confirmed results from this previous study that lysate applied via perfusion can drive a transient contraction in *Xenopus* embryonic epithelia that lasts ~2 min (Fig. 1A). Contraction begins 10–20 s after lysate is applied and continues for 30–90 s. Repeated perfusion of the same embryo at 30 s intervals demonstrates that the contraction response saturates (Fig. 1B; Movie 1), whereas pulsed delivery of lysate at 600 s intervals reveals that the response does not desensitize (Fig. 1C; Movie 2).

### The contraction response is developmentally regulated

We next mapped temporal and spatial responses of developing embryos to lysate. We perfused different regions of developing embryos from late blastula (stage 9) to neural plate stages (stage 17). Animal cap ectoderm, the marginal zone, and all tissues near or within the neural plate epithelium respond; however, vegetal endoderm does not, even as nearby marginal zone cells are observed to contract (Fig. 1D). Next, we perfused lysate onto animal cap ectoderm in whole embryos ranging from early blastula to early neurula stages and found that early blastula-stage epithelia would not contract but that animal cap ectoderm and epidermis at later stages were fully responsive (Fig. 1E). The limited stage- and tissue-specific responses suggest that the molecular pathway mediating the contractile response is developmentally regulated.

### Lysate activates actomyosin contractility

Actomyosin contractility within the medio-apical cortex and apical junctions play a key role in epithelial homeostasis and morphogenesis (Agarwal and Zaidel-Bar, 2018; Blanchard et al., 2018; Martin et al., 2010; Miao and Blankenship, 2020; Sawyer et al., 2010). We previously found that both the apical and basal cell cortex transiently remodel F-actin during exposure to lysate (Joshi et al., 2010). To test whether F-actin and non-muscle myosin II mediate the epithelial contractile response to lysate, we used a panel of small molecules to inhibit actin polymerization (0.6 µM latrunculin B, LatB) (Benink and Bement, 2005; Lee and Harland, 2007), to inhibit Rho kinases (50 µM Y-27632) (Maekawa et al., 1999; Narumiya et al., 2000) and to inhibit non-muscle myosin II heavy chain (100 µM blebbistatin, BBS) (Lee and Harland, 2007; Straight et al., 2003). We incubated early gastrula-stage embryos for 30–60 min in each inhibitor, and perfused cell lysate over the animal ectoderm (Fig. 2A; Movie 3). Kymographs collected transverse to the flow stream at the center of the contractile region (Fig. 2A′) reveal both qualitative and quantitative changes in contraction magnitude and kinematics of the contraction (see Fig. 2B for definitions of contraction strength, delay, time-to-peak and duration; Joshi et al., 2010; Kim et al., 2014). Notably, the

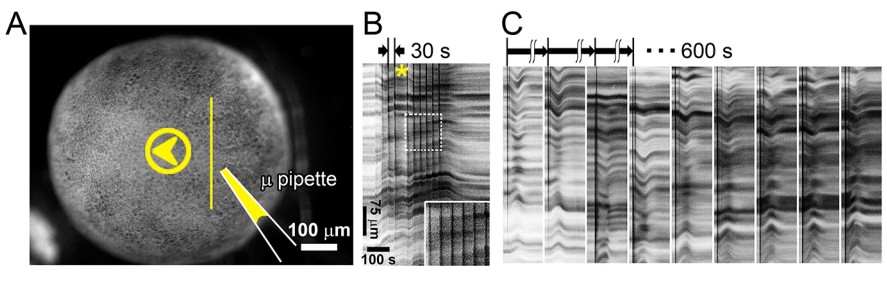

**Fig. 1. Spatiotemporal responses to lysate suggest that lysate sensing does not desensitize and is developmentally regulated**. (A) A single frame from a time-lapse sequence demonstrating multiple pulses of lysate to the ectodermal surface of a gastrula-stage embryo. Arrowhead (yellow) shows the direction of lysate flow from the micropipette (μ pipette). Vertical line indicates the spatial domain along which kymographs are collected. (B) Kymograph from the time-lapse sequence of ten contractions in the same embryo (with a single 60 s gap, asterisk; Movie 1). The surface is perfused with pulses of 60 nl lysate every 30 s. The embryonic epithelium recoils after the pulse train ends. Inset (dashed box) shows little change after multiple pulses. Data shown are representative of nine embryos. (C) Representative kymographs showing the responses of a single embryo perfused nine times with 60 nl lysate every 600 s. Each kymograph is the same scale as in B and starts 600 s (10 min) after the start of the previous one. The full set of nine perfusions takes place over ~90 min. The ectoderm contracts and relaxes after each perfusion (Movie 2; nine sequences are concatenated). Data shown are representative of three embryos. (D) Contractile responses to perfused lysate are observed in all epithelial tissues in gastrula and neurula stages, except vegetal endoderm (st. 10–11, gastrula stages). (E) Contractile responses in ectoderm perfused with lysate begin in late blastula stages and continue through neurulation.

incidence of contractions is the same after inhibition of F-actin polymerization or Rho kinases, but is reduced or eliminated when myosin II is inhibited with BBS (Fig. 2C). In concordance with the role of actomyosin in cell contractility, all treatments significantly reduced the strength, or absolute magnitude, of the contractile response (Fig. 2D). BBS produced the largest reduction in contraction strength and increased the delay between stimulation and onset of the contractile response in cases where a contraction was observed (Fig. 2E). Although BBS did not significantly reduce the time-to-peak (Fig. 2F), the inhibition of myosin II contractility shortened the duration of contractions (Fig. 2G). Inhibition of F-actin polymerization and inhibition of Rho kinases both significantly reduced contraction strength (Fig. 2D) but were less effective than direct myosin II inhibition. Remarkably, disruption of F-actin by LatB to the point where epithelial integrity is lost and cell–cell junctions begin to rupture (see asterisks in Fig. 2A) only moderately reduced contraction strength. Thus, actomyosin contractility is a key target of the cellular response to acute lysate stimulation.

## Extracellular nucleotides ATP, ADP, UTP and UDP, present in lysate, induce contractions

We tested several nucleotides and other candidate compounds to identify what factor or group of factors induced actomyosin contractility. Using a perfusion-based candidate screen (Table 1), we confirmed eATP as a contractility agonist (Kim et al., 2014) and that extracellular UTP is also capable of inducing contractions. Extracellular ADP, adenosine (ADO) and UDP also induce contractility but only at concentrations of 40 μM or higher. Additionally, the energy source of eATP is not required to drive contractions since ATPγS, a non-hydrolysable analog of ATP, can

also drive contractility. Treatment of ATP with shrimp alkaline phosphatase (SAP), a nucleotide-hydrolyzing enzyme, abrogated the ability of 40 μM ATP to induce contractions but did not reduce contractions driven by ATPγS. To test whether eATP or other nucleotides in lysate were triggering contractility, we incubated lysate with SAP and found it could not induce contractions, similar to embryo culture medium, salts, sodium glutamate and acetylcholine (Table S1). Thus, eATP and a limited set of other nucleotides present in lysate are sufficient to induce contractions with relative selectivity (ATP≈UTP>ADP≈UDP>ADO). Since ATP is by far the most abundant nucleotide in *Xenopus* egg cytoplasm, at concentrations of ~1 mM (Woodland and Pestell, 1972), we focused this study on the effects of eATP.

## The purinergic receptors mediate effects of eATP on contractility

eATP is sensed by cells via two families of purinergic cell surface receptors, GPCRs of the P2Y family (P2RY) and L-type $Ca^{2+}$-channel-coupled receptors of the P2X family (P2RX) (Abbracchio and Burnstock, 1994; Giuliani et al., 2019). Both receptor families elevate intracellular $Ca^{2+}$ concentration in response to eATP. In *Xenopus*, eATP can trigger $Ca^{2+}$ influx in embryonic ectoderm (Kim et al., 2014) as well as bursts of F-actin assembly in the apical cell cortex (Arnold et al., 2019). To test the potential role of P2RX-coupled $Ca^{2+}$ channels, we incubated embryos in the L-type $Ca^{2+}$ channel antagonist lanthanum chloride (4 mM) (Lettvin et al., 1964), perfused lysate and found that ectoderm contracted with the same incidence and strength as controls (Table S2). To further discriminate between the potential roles of P2RY and P2RX eATP receptors, we perfused lysate onto embryos incubated in a competitive inhibitor of P2RX, pyridoxal phosphate-6-azophenyl-2′,4′-disulfonate (PPADS, 100 μM; Table S2)

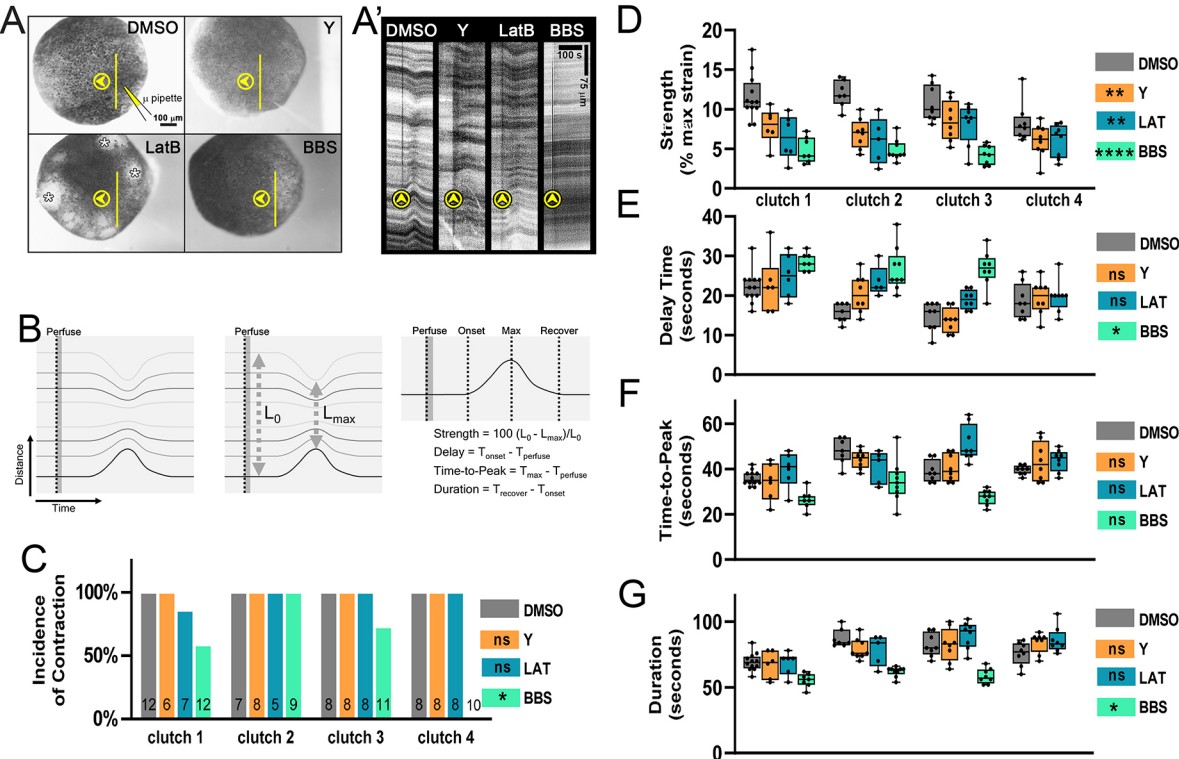

**Fig. 2. F-actin and myosin II are involved in the contractile response to ATP.** (A) Single time-lapse frames of embryos incubated in DMSO (carrier control), Y-27632 (Y), latrunculin B (LatB) or blebbistatin (BBS) prior to being exposed to a single 60 nl pulse of lysate. Arrowheads indicate direction of perfusion flow from the micropipette (μ pipette, position indicated for DMSO; see Movie 3). (A′) Kymographs collected from time-lapse sequences along the lines marked in A show contractions for each condition. Vertical arrowheads indicate the time of perfusion. Multiple lesions in the epithelium appear after treatment with LatB (asterisks in A). (B) Schematic kymograph showing a mock contraction with lines indicating pigment features displacing over time and their analysis under each condition. (C) Incidence of contraction (expressed as a percentage), (D) strength, (E) delay time, (F) time-to-peak and (G) duration of the contraction in the indicated conditions. n embryos is indicated in C. Note: cases where contractions were absent are not included in D–G. Box plots in D–G show the median (line), interquartile range (box) and range (whiskers), along with individual data points. Statistics for C–G indicate significance from the DMSO control condition. ns, not significant; *P<0.05; **P<0.01; ****P<0.0001 (nested one-way ANOVA with Dunnett's multiple comparisons test in C, Šidák's multiple comparisons test in D–G).

(Lambrecht et al., 1992; Ziganshin et al., 1993). When perfused with high concentrations of eATP (400 μM), ectoderm continued to contract; however, at lower eATP concentrations (0.4 μM), PPADS eliminated contractions. Since inhibition of P2RX receptors failed to block eATP-induced contractility, we focused our candidate screen on the P2RY family of receptors.

### Purinergic receptor P2RY2 mediates the contractility response to eATP

The *X. laevis* genome (Session et al., 2016) encodes 11 members of the P2RY family. To narrow the set of candidates, we used reverse transcription PCR (RT-PCR) to screen expression of P2RY receptors that shared ligand specificity (ATP, UTP, ADP, UDP or ADO) (Table S3) to determine which receptors are expressed at the stages or in tissues with visible contractile responses (Fig. 1C,D). Additionally, we ruled out roles for two P2RY family members, P2RY8 and P2RY11, that had been previously described. One family member, P2RY8, appears to be associated with *Xenopus* neural patterning (Bogdanov et al., 1997) but is not expressed in the late blastula or early gastrula. Another family member, P2RY11, has been associated with sporadic contraction (Shindo et al., 2010) and cyclic adenosine monophosphate (cAMP) signaling (Devader et al., 2007) during *Xenopus* gastrulation, but this receptor was not reported to be expressed in animal ectoderm (Devader et al., 2017). Of the remaining candidates, we found that P2RY2 homoeologs P2RY2.L and P2RY2.S are expressed in the early embryo with a

temporal pattern that closely aligns with gastrulation (Session et al., 2016). We further confirmed the expression of P2RY2 by RNA *in situ* hybridization and found that it is absent in early- and mid-gastrula endoderm but expressed in ectoderm (Fig. 3A), consistent with the spatial pattern observed in perfusion studies (Fig. 1D).

We tested the necessity of P2RY2 in eATP-induced contractility by simultaneous knockdown of both P2RY2 homoeologs. We co-injected two morpholino oligonucleotides designed to block the translation initiation sites of P2RY2.L and P2RY2.S (P2RY2-MO) into animal blastomeres at the eight-cell stage (32 ng total/embryo). We found that P2RY2-MO strongly inhibits contraction responses to eATP perfusion, with only 25% of P2RY2-MO-injected embryos exhibiting a contractile response compared to 93% of control morpholino (CO-MO)-injected embryos (Fig. 3B,C). Furthermore, co-injection of a morpholino-resistant mRNA encoding P2RY2.L successfully rescued eATP-induced contractility in P2RY2-MO-injected embryos, with 88% of embryos responding to eATP.

### Purinergic receptor P2RY2 regulates apical F-actin levels, increasing levels after eATP perfusion

Perfused eATP and lysate can drive rapid remodeling of the actin cytoskeleton, enhancing apical networks (Arnold et al., 2019) and transiently depolymerizing and remodeling basal networks (Joshi et al., 2010). To test whether apical F-actin assembly is regulated by P2RY2, we injected CO-MO or P2RY2-MO contralaterally

Journal of Cell Science

**Table 1. Testing nucleotides and nucleosides for ability to induce epithelial contraction**

| Candidate factor | Concentration | Number of embryos | Incidence of contraction (%)[1] | Contraction magnitude[2] |
|---|---|---|---|---|
| ATP | 400 µM | 6 | 100 | ++ |
| | 40 µM | 6 | 100 | ++ |
| | 0.4 µM | 18 | 72 | + |
| ADP | 40 µM | 4 | 100 | + |
| ADO | 400 µM | 8 | 75 | + |
| | 40 µM | 5 | 0 | – |
| cAMP | 40 µM | 5 | 0 | – |
| CTP | 40 µM | 8 | 0 | – |
| GTP | 40 µM | 6 | 0 | – |
| UTP | 40 µM | 6 | 100 | ++ |
| UDP | 40 µM | 6 | 100 | + |
| | 0.4 µM | 5 | 0 | – |
| TTP | 40 µM | 8 | 0 | – |
| ATPγS | 2 mM | 14 | 100 | + |
| | 1 mM | 7 | 100 | + |
| | 100 µM | 13 | 15 | + |
| SAP-treated ATP | 40 µM | 6 | 0 | – |
| SAP-treated ATPγS | 1 mM | 10 | 100 | + |

[1]Incidence of contraction includes contractions of any magnitude (++ or +) in response to perfusion. [2]Contraction magnitude is qualitatively assessed by comparing nucleoside/nucleotide perfusions of embryos with an observable contraction. Contraction magnitude is categorized with respect to 40 µM ATP or lysate: ++, contraction is equal or stronger; +, contraction is weaker; –, contraction not observed.

together with a Rhodamine–dextran lineage label into the animal pole at four- or eight-cell stages. We raised the injected embryos to early gastrula stages, explanted animal caps onto fibronectin-coated substrates (Joshi and Davidson, 2010), and added ATP or culture medium to the samples. After 30 min, explants were fixed and stained with phallacidin to quantify F-actin intensity in both injected and uninjected cells. Due to sample-to-sample variation in fixation and staining, we normalized levels of F-actin to levels in neighboring uninjected cells (Fig. 3D,E). In the absence of ATP, ectoderm cells expressing P2RY2-MO showed 11% reduction in F-actin compared to uninjected cells in the same explant. The impact of the P2RY2 knockdown was pronounced after P2RY2-MO-expressing cells were exposed to eATP, resulting in a 40% reduction in F-actin intensity in P2RY2-MO-expressing cells, as compared to uninjected cells. The CO-MO-injected cells also demonstrated a 10% reduction in F-actin compared to uninjected cells. Together with the ability of actomyosin inhibitors to modulate contractile responses to lysate, our knockdown findings support the central role of P2RY2 in eATP-induced contractility via F-actin remodeling.

Once activated by eATP, P2RY2 might drive contractility through direct association with the cytoskeleton (Haenig et al., 2020; Orchard et al., 2014; Yu et al., 2008) or via G protein activation of downstream actomyosin effectors such as RhoA or phospholipase C (Erb and Weisman, 2012; Flock et al., 2017). To test the role of G protein signal transduction pathways, we co-injected mRNA encoding prenylation-deficient forms of the heterotrimeric G protein γ subunits Gγ-3 (GNG3) and Gγ-5 (GNG5) in which the prenylation sequence CAAX has been mutated to SAAX (Mulligan et al., 2010). SAAX forms of Gγ proteins bind Gβ proteins, sequestering them in the cytoplasm, and inhibit Gβγ localization and activation by GPCRs at the plasma membrane. Animal cap epithelia expressing both Gγ-3-SAAX and Gγ-5-SAAX proteins from injected mRNAs exhibited 40–60% of the contractility strength of controls (Fig. S1A,B). The reduction of contractility by Gγ-SAAX overexpression demonstrates involvement of a G-protein-mediated pathway.

Studies have identified long-range signaling pathways for sensing and activating the innate immune response in the zebrafish larval tail and for regeneration in planarians. In wounded zebrafish larva, osmotic shock at the wound site (Enyedi et al., 2013), ATP release (Gault et al., 2014), production of reactive oxygen species such as hydrogen peroxide ($H_2O_2$) (Niethammer et al., 2009), and lipid peroxidation and production of arachidonate metabolites (Enyedi et al., 2016; Katikaneni et al., 2020) have been found to guide leukocyte infiltration to wound sites. Injury in the planarian also drives rapid activation of extracellular signal-regulated kinase (Erk) across the entire body of the flatworm. Propagation of a wave of Erk activation reaches ~1 mm per hour. This rate is similar to the 7 mm per hour propagation of contractility in response to microfluidically streamed eATP seen previously by our group (Kim et al., 2014), albeit over a shorter range of 100 µm. Contractile responses to eATP or lysate might reflect a response to either lytic wounding or sublytic cell stress. Our previous work indicates that a range of sublytic stimuli, including laser activation, electrical stimulation and lysate, all produce temporally similar contractile responses (Joshi et al., 2010). However, the pathways activated downstream of osmotic shock in the zebrafish tail wound, including $H_2O_2$ and arachidonic acid release, trigger the innate immune response and recruit leukocytes to the wound area but no do not appear to trigger contraction at the wound site. We note that the innate immune system in frogs is not yet developed at the embryonic stage used in our studies.

Extracellular nucleotides, including ATP and UTP released from wounded epithelia, can regulate actomyosin contractility. Here, we demonstrate that diffusible eATP drives contractility in embryonic epithelia through P2RY2. eATP and its cellular receptor P2RY2 are well-known regulators of embryonic and adult tissue mechanics (Burnstock and Verkhratsky, 2012; Mikolajewicz et al., 2019; Verkhratsky and Burnstock, 2014). eATP and P2RY2 are involved in mammalian decidualization and implantation (Gu et al., 2020). eATP release after mechanical stimulation is well characterized in the cardiovascular system, where it plays a role in flow sensing by endothelial (Milner et al., 1990) and red blood cells (Wan et al., 2011), triggering valvulogenesis (Fukui et al., 2021), and is involved in the cellular stress response (Galluzzi et al., 2018). eATP and P2RY2 additionally regulate tissue responses to mechanical stimulation of lung epithelia (Homolya et al., 2000), cancer (Burnstock and Di Virgilio, 2013) and the central nervous system (Burnstock, 2008). The activity of eATP and P2RY2 suggests a role in *Xenopus* epithelial homeostasis and morphogenesis. Our results demonstrate that actomyosin contractility, likely driven by eATP, could be a key mechanism for controlling the contractile forces that shape tissues and drive organogenesis. This suggests a linkage between biochemical regulation of actomyosin and supracellular control of tissue-scale movements.

## MATERIALS AND METHODS
### Embryo and explant culture, microsurgery, histology, morpholinos and RNA *in situ* hybridization
Eggs were obtained from female *X. laevis* frogs, fertilized, dejellied in 2% cysteine solution (pH 8.4) and cultured in 1/3× Modified Barth's solution (MBS) following standard methods (Kay and Peng, 1991) in accordance with IACUC animal use protocols of the University of Pittsburgh. Embryos were staged appropriately (Nieuwkoop and Faber, 1967), and vitelline membranes were removed using forceps. Animal cap explants were

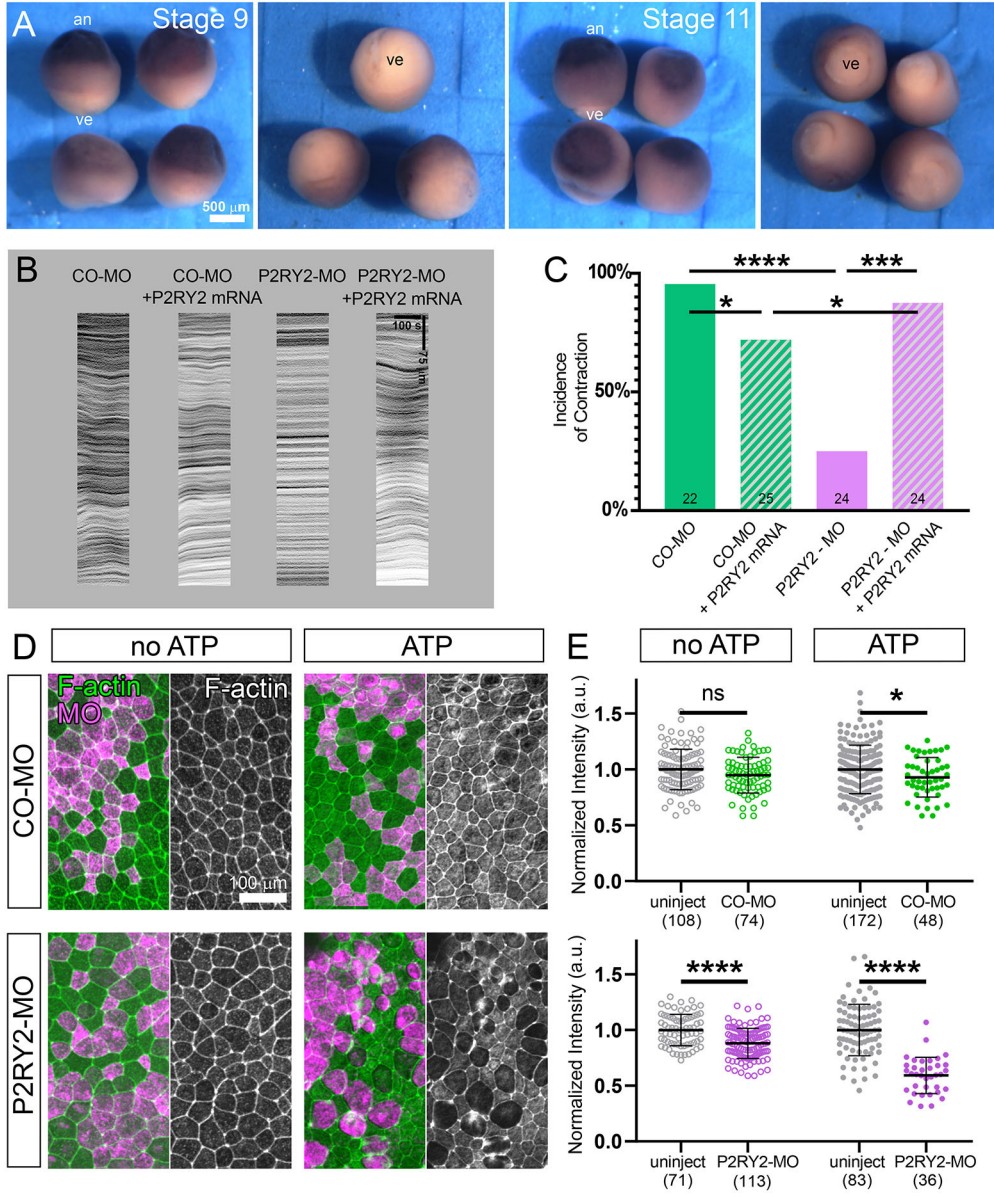

**Fig. 3. Gene expression and knockdown of the P2RY2 receptor.** (A) RNA *in situ* hybridization gene expression of P2RY2 at late blastula (stage 9) and mid-gastrula (stage 11). Highest expression is seen in the animal pole (an) with the lowest in the vegetal endoderm (ve). Data shown are representative of three clutches. (B) Representative kymographs of contractility of morpholino controls, knockdowns and rescues after perfusion with 4 nl of 400 µM ATP. For knockdown of P2RY2, embryos were injected with either 32 ng of a control morpholino oligomer (CO-MO) or 16 ng each of two morpholino oligomers targeting translation initiation sites of the two homeoalleles, P2RY2.L and P2RY2.S (P2RY2-MO). Rescue was tested in embryos co-injected with 2 ng of a morpholino-resistant P2RY2.L mRNA. (C) The incidence of contractility for knockdown and rescue experiments. *n* embryos is indicated. (D) P2RY2 knockdown effects on actomyosin accumulation after exposure to eATP. The left side of each panel shows F-actin staining (green) and a scattered population of cells injected with the morpholino and a Rhodamine–dextran tracer (magenta). The right side of each panel shows a grayscale image of the F-actin channel. (E) Medio-apical F-actin intensity was measured through mosaic knockdown and control morpholino-injected explants and compared to uninjected cells. The intensity of each region of interest, in both uninjected and morpholino-injected cells, was normalized to the mean F-actin intensity of the uninjected cells in the same explant. Results for each condition are pooled from 3–4 explants, and cell numbers are listed below each data column. Mean±s.d. is indicated. a.u., arbitrary units. Statistics for C and E indicate significance of the incidence of contraction and the normalized intensity of F-actin, respectively. ns, not significant; *$P<0.05$; ***$P<0.001$; ****$P<0.0001$ (two-way ANOVA in C; one-tailed unpaired or Welch's *t*-test in E).

microsurgically isolated and cultured in Danilchik's For Amy solution (DFA; Sater et al., 1993) on fibronectin-coated glass substrates (Joshi, 2011). Explants processed for F-actin localization were fixed with 4% paraformaldehyde and 0.25% glutaraldehyde in PBST (1× PBS with 0.1% Triton X-100) for 15 min at room temperature. After washing, the samples were incubated with BODIPY-FL–phallacidin (1:800; Thermo Fisher, Waltham, MA, USA) to visualize F-actin.

Morpholino oligonucleotides were designed with GeneTools Inc. (Philomath, OR, USA) to block the translational start sites: P2RY2.L,

5′-CTTGGTCTCCAGACAAATTCATTTT-3′; P2RY2.S, 5′-TTCTGGGTC-TTCAAACACATTCATC-3′ (CO-MO, 5′-CCTCTTACCTCAGTTACAA-TTTATA-3′). For mosaic labeling, blastomere derivatives injected with morpholinos were co-injected with lysine-fixable 70 kDa tetramethylrhoda-mine dextran (Thermo Fisher, Waltham, MA, USA).

P2RY2 is a good candidate for morpholino-based knockdown as its two homoeologs are not expressed until after the maternal–zygotic transition (Session et al., 2016) and maternal contributions of P2RY2 are not observed in proteomic studies of the egg (Wühr et al., 2014). Although morpholinos

can activate the innate immune system (Gentsch et al., 2018), our studies are carried out in the early embryo, well before there is an active immune response (Paraiso et al., 2019).

We note that the coding region of both P2RY2 homoeologs consist of a single exon, precluding design of splice acceptor morpholinos. We used BLAST to check for potential off-target binding sites of the morpholinos. In the *X. laevis* genome, the only identified binding sites were the targeted translation initiation sites of P2RY2.L and P2RY2.S. The only additional sequences had weak homology and are unlikely to be bound by the morpholinos; moreover, these additional sequences are tens of kb away from any other gene. Searching the *X. laevis* transcriptome also did not uncover any additional likely binding sequences in other transcripts.

Embryos were processed for gene expression analysis using standard RNA *in situ* hybridization protocols (Harland, 1991). Probe cDNAs were amplified from stage 10 libraries with the following primers: P2RY2.L forward primer, 5′-GGTTGAGACTACAAAACCAGAA-3′; P2RY2.L reverse primer, 5′-CTATGTTACCTCATGGAGGTTG-3′; P2RY2.S forward primer, 5′-GGTTAAAACAACAAAACGAGAAAG-3′; P2RY2.S reverse primer, 5′-GGTATATGTAGATTTACCTACAGCC-3′.

Plasmids encoding *in situ* hybridization probes and rescue mRNA for P2RY2 are available upon request. Plasmids encoding prenylation-deficient G protein γ subunits were a kind gift from Steve Farber. Plasmids were sequence verified and linearized with appropriate restriction enzymes (New England Biolabs, Ipswich, MA, USA), purified (Zymo Research, Irvine, CA, USA) and *in vitro* transcribed (CellScript, Thermo Fisher, Waltham, MA, USA). RNA was purified (Zymo Research) and analyzed by gel electrophoresis before injecting.

cDNA libraries were made from total embryonic RNA from embryos at the indicated stages by reverse-transcription (SuperScript, Thermo Fisher, Waltham, MA, USA) and a non-specific oligo-dT primer. Libraries were interrogated for presence of the indicated P2RY family using the primers in Table S3 (Primer 1 and Primer 2). Products were scored for presence of the respective band by agarose gel electrophoresis.

## Lysate, ATP and compound perfusion
The perfusion and lysate preparation methods have been described previously (Joshi et al., 2010). Lysate and dilute perfusion compounds were freshly prepared on the day of the experiment. Salts, bioactive compounds, nucleotides and nucleosides were diluted in 1/3× MBS (see Table 1 and Table S1 for tested compounds). All compounds were stored and utilized according to the product guidelines (Sigma-Aldrich, St. Louis, MO, USA). To visualize cell lysate and other perfused solutions, 3 µl black non-waterproof ink (Higgins Fountain Pen India Ink; Utrecht Art, Cranbury, NJ, USA) was added to each 100 µl stock perfusate. Since contractile responses are variable in embryos from one mating to the next (i.e. 'clutch') – for instance, the strength of contraction varies from 9% to 12% – we compare results from multiple clutches. Lysate and nucleotides were incubated with 10 U SAP per 100 µl for 1 hour before use.

## Image acquisition, analysis and statistical analysis
Still images, time-lapse sequences of developing embryos and time-lapse sequences of perfusion experiments were recorded as described previously (Joshi and Davidson, 2010) using a CCD camera (Scion Corp., Frederick, MD, USA) mounted on a dissecting stereomicroscope. Image acquisition and subsequent analysis of time-lapse sequences were carried out using custom-written macros and plug-ins for image analysis software [MicroManager 1.4 and 2.0 (Edelstein et al., 2014); ImageJ version 1.54 (Schneider et al., 2012)]. To generate kymographs, a line region-of-interest spanning the center of the embryo was selected and resliced over the full time-lapse sequence. Contrast was enhanced, and each kymograph was manually categorized for incidence into strong contraction, weak contraction or no contraction categories. Measurements of contraction strength, delay time, time-to-peak and duration were quantified as previously described (Joshi et al., 2010). Tests for statistical significance of treatments were determined with simple and paired *t*-tests, and one-way ANOVAs, two-way ANOVAs or Mann–Whitney *U*-tests (Sokal and Rohlf, 1987) using commercial statistical software (GraphPad Prism v. 10 and SPSS v. 25). Whiskers indicate either minimum to maximum values (Fig. 2D–G) or ±s.d. (Fig. 3E).

## Acknowledgements
We would like to thank members of the MechMorpho Lab for their support and persistence as this project evolved ("Logic is the beginning of wisdom, not the end" – Spock, *Star Tek VI: The Undiscovered Country*). We also thank Steve Farber for sharing plasmids.

## Competing interests
The authors declare no competing or financial interests.

## Author contributions
Conceptualization: S.D.J., L.A.D.; Formal analysis: L.A.D.; Funding acquisition: L.A.D.; Investigation: S.D.J., T.R.J., L.Z., C.S.; Methodology: S.D.J., T.R.J., L.Z., L.A.D.; Project administration: L.A.D.; Resources: L.Z., C.S.; Supervision: L.A.D.; Validation: C.S.; Writing – original draft: S.D.J., T.R.J., L.Z., C.S., L.A.D.; Writing – review & editing: S.D.J., T.R.J., L.Z., C.S., L.A.D.

## Funding
This work was supported by grants from the National Institutes of Health (NIH; R01 HD044750 and R37 HD044750) and the Swanson School of Engineering, University of Pittsburgh. T.R.J. was additionally supported by the Cardiovascular Bioengineering Training Program (NIH National Heart, Lung, and Blood Institute T32 HL076124). Open Access funding provided by University of Pittsburgh. Deposited in PMC for immediate release.

## Data and resource availability
All relevant data and details of resources can be found within the article and its supplementary information.

## Peer review history
The peer review history is available online at https://journals.biologists.com/jcs/lookup/doi/10.1242/jcs.263877.reviewer-comments.pdf

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
