## [Peer Review File · Journal of Cell Science]

Supracellular contractility in *Xenopus* embryo epithelia regulated by extracellular ATP and the purinergic receptor P2Y2

Sagar D. Joshi, Timothy R. Jackson, Lin Zhang, Carsten Stuckenholtz and Lance A. Davidson
DOI: 10.1242/jcs.263877

Editor: Kathleen Green

Review timeline

Original submission:	24 January 2025
Editorial decision:	17 February 2025
First revision received:	12 May 2025
Editorial decision:	21 May 2025
Second revision received:	12 August 2025
Accepted:	18 August 2025

Original submission

First decision letter

MS ID#: jcs.263877

MS TITLE: Supracellular contractility in *Xenopus laevis* embryonic epithelia regulated by extracellular nucleotides and the purinergic G-protein coupled receptor P2Y2

AUTHORS: Sagar D Joshi; Timothy R Jackson; Lin Zhang; Carsten Stuckenholtz; Lance Davidson

ARTICLE TYPE: Short Report

Dear Dr Davidson,

We have now reached a decision on the above manuscript.

To see the reviewers' reports and a copy of this decision letter, please go to:

As you will see, the reviewers raise a number of substantial criticisms that prevent me from accepting the paper at this stage. They suggest, however, that a revised version might prove acceptable, if you can address their concerns. If you think that you can deal satisfactorily with the criticisms on revision, I would be pleased to see a revised manuscript. We would then return it to the reviewers.

Reviewer 1

Advance summary and potential significance to field

The manuscript follows from a previous publication from the authors (Joshi et al, Exp Cell Res 316(1): 103-14 (2010)), although there has been a considerable time delay. This previous publication showed, amongst other things, that *Xenopus* gastrulae extracts induced actinomycin-dependant contraction when perfused onto epithelia of *Xenopus* gastrulae. In the current manuscript they follow up this observation to show that extracellular nucleotides (e.g. ATP and UTP) can generate the same response when perfused onto *Xenopus* gastrulae. They show that this response is only evident from late blastulae, but is then widespread within the embryo (except for

the vegetal pole (endoderm) of gastrulae. The authors then inhibit the actinomyosin contractile apparatus, using small molecule inhibitors, and show that the contractile response to extracellular ATP (eATP) is reduced and in one case completely inhibited. The data is convincing enough to suggest that actinomyosin is required for the contractile response to ATP.

In mammalian systems, responses to eATP are known to be mediated through two families of cell surface receptors; L-type calcium channels (P2X receptors) and G-protein coupled receptors (P2Y receptors). P2X inhibitors have very little effect on the responses in this study suggesting that they might be mediated by P2Y receptors. Of the candidate receptors only P2Y2 had the correct expression pattern. The authors injected antisense morpholino oligonucleotides for P2Y2 (P2YR2-MO) into the animal hemisphere of *Xenopus* embryos at the 8-cell stage and then tested the response of gastrula animal epithelium to perfusion of eATP. The authors found that the response was reduced when compared to embryos injected with control morpholino. The response was rescued when P2YR2-MO was coinjected with an morpholino resistant mRNA for P2Y2. This is highly suggestive that the contractile response of *Xenopus* epithelia to eATP is mediated by P2YR2. Consistent with this, the authors also showed that G-protein signal transduction pathways are also required.

The authors do a good job in showing that the eATP-P2YR2 signalling pathway regulates actinomyosin and epithelial contraction in *Xenopus* embryos, an observation that maybe more widely applicable - to wound healing for example.

Comments for the author

While I tend to believe the data, one issue with the morpholino experiments is that the authors don't demonstrate that the P2YR2-MOs actually inhibit translation of P2YR2 mRNA. This is rather difficult without a sensitive antibody to P2YR2 protein, but the authors could show that it inhibits coinjected mRNA. They could also then show that the MO resistant P2YR2 mRNA is actually resistant. However, the rescue experiment they perform with the MO resistant P2YR2 mRNA is considered an adequate control by many colleagues. It is just that you can't tell whether it is actually rescuing the signal pathway or compensating for its absence. An alternative strategy used by many colleagues is to show the same result with a second set of P2YR2-MOs.

The manuscript is generally well written and is easily understood, but the last sentence on page 4 needs to be written more clearly.

Be consistent with the nomenclature that you use for P2Y receptors. You write about P2Y8 and P2Y11 but P2YR2. I believe that the latter is the currently accepted way of naming these receptors so it should also be P2YR8 and P2YR11.

Reviewer 2

Advance summary and potential significance to field

Comments for the author

Please reference the appropriate Niethammer papers.

The authors observe that cell lysate triggers a contractile response in *Xenopus* embryo epithelia, which this reviewer finds interesting. Then they go on to show that this response is mediated, at least in part, by extracellular nucleotide triphosphates (eATP and eUTP) binding to P2YR, a GPCR. The experimental data identifying both ligand and receptor are convincing, though they do not rule out the existence of parallel pathways. There has been very little work on this receptor in embryos, and relatively little on its eATP ligand. Thus, the paper provides interesting new information on both and suggests that further exploration of eATP biology in early vertebrate embryos will be interesting. This is nice physiology and suited for the journal.

This reviewer disagrees with the authors main conclusion, that eATP binding to P2YR mediates normal morphogenesis. Based in the data provided and literature, including a paper the authors need to cite, this pathway is more likely to mediate a wound healing response (see below). Testing whether P2YR knockdown affects normal development or/and wound healing would strengthen the paper by giving it a biological result, even if both experiments came back negative.

Detailed comments:

The intro starts: *“While there is widespread belief that cell and tissue mechanical processes operating during vertebrate development”* and in the results the authors state *“The activity of eATP and P2RY2 suggest a role in Xenopus epithelial homeostasis and morphogenesis. The results demonstrate actomyosin contractility, likely driven by eATP could be a key mechanism for controlling the contractile forces that shape tissues and drive organogenesis”*.

In these statements, the authors seem to conclude that the eATP-P2YR pathway plays a role in normal development. No data are provided in support of this hypothesis, eg no development defects are reported in the receptor knockdown experiment. This reviewer suspects that a pathway which responds to cell lysate and eATP is more likely to be a wound response pathway than a normal developmental pathway. Wound response pathways are essential for organismal fitness and of great interest from a pathology perspective. It seems likely that in the wild, frog embryos would often suffer wounding by predators and therefore a wound response pathway would increase fitness. I suggest normal development and wound healing be given equal weight in the intro and discussion. Better, the authors should test both experimentally.

The authors should read and cite the work of Philip Niethammer’s group on contractile response to eATP in embryonic epithelia in zebrafish (PMID: 25533845 and possibly also PMID: 32521230). The first paper at least is directly relevant to this submission. There is no direct overlap, rather this paper may have identified the receptor that was implied but not identified in Niethammer’s work. Citing that work appropriately would force the authors to consider the hypothesis that they are working on a pathway that mediate embryo wound responses rather than assume it mediates normal development.

The paper would be more interesting if the authors tested whether P2Y2 plays a role in normal development and/or response to mechanical wounding. They have a knockdown method, so it should not be too hard to score effects on embryogenesis. Effects of a knockdown on wound healing might be a bit harder to measure, but others have scored this for different molecular perturbations. Presumably they could deliver a somewhat standardized wound with a needle or laser, then quantify recovery kinetics or survival with P2YR expression intact or knocked down.

First revision

Author response to reviewers' comments

We thank the editor and reviewers for their careful evaluation and constructive comments. We acknowledge this project began quite some time ago, but the findings remain highly relevant.

Before we address the individual reviewers' comments we would like to comment on the goal and scope of this manuscript. As noted, we previously discovered the role of cell lysate during a study of epithelial wounding. This manuscript follows up on this work by seeking to identify the cell surface receptors responsible for transducing the lysate stimulus into an epithelial contraction. This manuscript was formatted as a short report to reflect this limited goal.

The summary suggested expanding this manuscript to include analysis of the role of P2RY2 in an endogenous physiological process. We are preparing a second manuscript that includes such an analysis. This new manuscript does not focus on wound healing but rather the role of P2RY2 on mechanical feedback during morphogenesis. This second manuscript proposes that extracellular ATP (eATP) and P2RY2 form a negative feedback circuit that regulates the rate of large-scale tissue

deformations during gastrulation. In the next manuscript we find that high rates of deformation drive local eATP release, P2RY2 activation, and tissue stiffening. eATP driven tissue stiffening then slows the rate of epiboly during gastrulation. Our current manuscript lays an important foundation for the next manuscript.

We include a point-by-point response to the reviewers' comments below. We have indicated our responses in blue font.

Reviewer #1:

#1-1: While I tend to believe the data, one issue with the morpholino experiments is that the authors don't demonstrate that the P2RY2-MOs actually inhibit translation of P2RY2 mRNA. This is rather difficult without a sensitive antibody to P2RY2 protein, but the authors could show that it inhibits coinjected mRNA. They could also then show that the MO resistant P2RY2 mRNA is actually resistant. However, the rescue experiment they perform with the MO resistant P2RY2 mRNA is considered an adequate control by many colleagues. It is just that you can't tell whether it is actually rescuing the signal pathway or compensating for its absence. An alternative strategy used by many colleagues is to show the same result with a second set of P2RY2-MOs.

We appreciate the reviewer's perspective on the use of morpholinos. We have carried out our knock-down studies according to *Xenopus* community standards that include the use of control morpholinos and rescue (Eisen and Smith, 2008). Experimental designs involving morpholinos have been heavily debated in the field (Blum et al., 2015), but morpholinos remain an important tool in models such as zebrafish and *Xenopus*. Furthermore, morpholinos have been extensively used to study *Xenopus* epithelial biology due to the ability of these tissues to differentiate in isolation, independent of dorsal-ventral or anterior-posterior patterning processes active elsewhere (Werner and Mitchell, 2013; Brooks and Wallingford, 2015; Walentek and Quigley, 2017). In some regards, P2RY2 is a good candidate for morpholino-based knock down studies as its two homeologs are not expressed until after the maternal-zygotic transition (Session et al., 2016) and maternal contributions of P2RY2 are not observed in proteomics of the egg (Wühr et al., 2014).

One point of caution that is frequently raised regards activation of the innate immune system (Gentsch et al., 2018). However, our studies are carried out in the early embryo, well before there is an active immune response (Paraiso et al., 2019).

Another point of caution is the potential for off-target effects. Using BLAST, we checked for potential off-target binding sites of the morpholinos. In the *Xenopus laevis* genome, the only additional sequences had weak homology and are unlikely to be bound by the morpholinos; moreover, they are tens of kb away from any other gene. Searching the *Xenopus laevis* transcriptome also did not uncover any additional likely binding sequences in other transcripts. The only identified binding sites were the targeted translation initiation sites of p2ry2.L and p2ry2.S.

Lastly, the reviewer is correct that we do not have an antibody to confirm reduction of P2RY2, but we have a very strong phenotypic readout, e.g., loss or reduction of eATP driven contractility and rescue of contractility after injection with a MO-resistant P2RY2 mRNA. Additionally, as noted by the reviewer, injection of dominant negative G-protein signaling constructs phenocopy P2RY2 MO-injected embryos.

Regarding use of a second set of morpholinos, the reviewer is referring to the use of splice-site targeting morpholinos. This is a perfectly reasonable recommendation, however, as we state in the methods section, neither P2RY2 homeologs have introns within the coding region. (Please see the below diagram for the gene structure of both homeologs of *Xenopus laevis* p2ry2, p2ry2.L and p2ry2.S.) While both genes do have introns, in both cases the introns are confined to the 5' UTR of the mRNA (orange boxes denote the UTR); the coding sequence (blue boxes) is uninterrupted by introns. Therefore, we designed translation blocking morpholinos that bind at the indicated AUG site and interfere with translation.

We have revised the methods section to emphasize the inability to target splice acceptor sites and a statement about the limitations of morpholinos.

#1-2: The manuscript is generally well written and is easily understood, but the last sentence on page 4 needs to be written more clearly.

We thank the reviewer for pointing out this gaffe. We have revised the sentence for clarity:

"Notably, contractility occurs at the same rate after inhibition of F-actin polymerization or Rho Kinase, but is frequently eliminated when myosin II is inhibited with BBS (Fig. 2C)."

#1-3: Be consistent with the nomenclature that you use for P2Y receptors. You write about P2Y8 and P2Y11 but P2RY2. I believe that the latter is the currently accepted way of naming these receptors so it should also be P2RY8 and P2RY11

We thank the reviewer for this. We have revised the nomenclature throughout the manuscript.

Reviewer #2:

#2-1 Please reference the appropriate Niethammer papers.

We thank the reviewer for reference to these interesting papers. We have added the following paragraph to the discussion to relate the findings of the current manuscript to these papers.

"Studies have identified long range signaling pathways for sensing and activating the innate immune response to wounding in the Zebrafish larval tail. This pathway is initiated with osmotic shock at the wound site (Enyedi et al., 2013), ATP release (Gault et al., 2014), production of reactive oxygen species such as hydrogen peroxide (H₂O₂) (Niethammer et al., 2009), and lipid peroxidation and production of arachidonate metabolites (Enyedi et al., 2016; Katikaneni et al., 2020) to guide leukocyte infiltration to wound sites. The long-range contractile response to ATP or lysate in this study may represent a response to both lytic wounding and sublytic cell stress. Our previous work indicates that a range of sublytic stimuli, including laser, electrical, and lysate all produce temporally similar contractile responses (Joshi et al., 2010). However, the pathways activated downstream of osmotic shock in the Zebrafish tail wound including H₂O₂ and AA release trigger the innate immune response and recruit leukocytes to the wound area but do not appear to trigger contraction at the wound site. We note that the innate immune system in frogs is not yet developed at the embryonic stage used in our studies."

References cited

Blum, M., De Robertis, E. M., Wallingford, J. B. and Niehrs, C. (2015) 'Morpholinos: Antisense and Sensibility', *Dev Cell* 35(2): 145-9.

Brooks, Eric R and Wallingford, John B (2015) *In vivo investigation of cilia structure and function using Xenopus Methods in cell biology*, vol. 127: Elsevier.

Eisen, Judith S and Smith, James C (2008) 'Controlling morpholino experiments: don't stop making antisense'.

- Enyedi, B., Jelcic, M. and Niethammer, P. (2016) 'The Cell Nucleus Serves as a Mechanotransducer of Tissue Damage-Induced Inflammation', *Cell* 165(5): 1160-1170.
- Enyedi, B., Kala, S., Nikolich-Zugich, T. and Niethammer, P. (2013) 'Tissue damage detection by osmotic surveillance', *Nat Cell Biol* 15(9): 1123-30.
- Gault, W. J., Enyedi, B. and Niethammer, P. (2014) 'Osmotic surveillance mediates rapid wound closure through nucleotide release', *Journal of Cell Biology* 207(6): 767-782.
- Gentsch, George E, Spruce, Thomas, Monteiro, Rita S, Owens, Nick DL, Martin, Stephen R and Smith, James C (2018) 'Innate immune response and off-target mis-splicing are common morpholino-induced side effects in *Xenopus*', *Developmental cell* 44(5): 597-610. e10.
- Joshi, S. D., von Dassow, M. and Davidson, L. A. (2010) 'Experimental control of excitable embryonic tissues: three stimuli induce rapid epithelial contraction', *Exp Cell Res* 316(1): 103-14.
- Katikaneni, A., Jelcic, M., Gerlach, G. F., Ma, Y., Overholtzer, M. and Niethammer, P. (2020) 'Lipid peroxidation regulates long-range wound detection through 5-lipoxygenase in zebrafish', *Nat Cell Biol* 22(9): 1049-1055.
- Niethammer, P., Grabher, C., Look, A. T. and Mitchison, T. J. (2009) 'A tissue-scale gradient of hydrogen peroxide mediates rapid wound detection in zebrafish', *Nature* 459(7249): 996-9.
- Paraiso, Kitt D, Blitz, Ira L, Zhou, Jeff J and Cho, Ken WY (2019) 'Morpholinos do not elicit an innate immune response during early *Xenopus* embryogenesis', *Developmental cell* 49(4): 643-650. e3.
- Session, A. M., Uno, Y., Kwon, T., Chapman, J. A., Toyoda, A., Takahashi, S., Fukui, A., Hikosaka, A., Suzuki, A., Kondo, M. et al. (2016) 'Genome evolution in the allotetraploid frog *Xenopus laevis*', *Nature* 538(7625): 336-343.
- Walentek, Peter and Quigley, Ian K (2017) 'What we can learn from a tadpole about ciliopathies and airway diseases: using systems biology in *Xenopus* to study cilia and mucociliary epithelia', *Genesis* 55(1-2): e23001.
- Werner, Michael E and Mitchell, Brian J (2013) Using *Xenopus* skin to study cilia development and function *Methods in enzymology*, vol. 525: Elsevier.
- Wühr, Martin, Freeman, Robert M, Presler, Marc, Horb, Marko E, Peshkin, Leonid, Gygi, Steven P and Kirschner, Marc W (2014) 'Deep proteomics of the *Xenopus laevis* egg using an mRNA-derived reference database', *Current Biology* 24(13): 1467-1475.

Second decision letter

MS ID#: jcs.263877R1

MS TITLE: Supracellular contractility in *Xenopus* embryo epithelia regulated by extracellular ATP and the purinergic receptor P2Y2

AUTHORS: Sagar D Joshi; Timothy R Jackson; Lin Zhang; Carsten Stuckenholz; Lance A Davidson

ARTICLE TYPE: Short Report

Dear Dr Davidson,

We have now reached a decision on the above manuscript.

To see the reviewers' reports and a copy of this decision letter, please go to:

It appears that an attached file that contained additional detailed comments from one of the referees was missed, as there is not a response to those comments. Before a decision is made, please consider and respond to those comments, and if necessary make any additional revisions.

Second revision

Author response to reviewers' comments

We thank the editor and reviewers for their careful evaluation and constructive comments. We acknowledge this project began quite some time ago, but the findings remain highly relevant.

Before we address the individual reviewers' comments we would like to comment on the goal and scope of this manuscript. As noted, we previously discovered the role of cell lysate during a study of epithelial wounding. This manuscript follows up on this work by seeking to identify the cell surface receptors responsible for transducing the lysate stimulus into an epithelial contraction. This manuscript was formatted as a short report to reflect this limited goal.

The summary suggested expanding this manuscript to include analysis of the role of P2RY2 in an endogenous physiological process such as wound healing or development. We are preparing a second manuscript that does not address the role of P2RY2 on wound healing but rather on mechanical feedback during morphogenesis. This second manuscript proposes that extracellular ATP (eATP) and P2RY2 form a negative feedback circuit that regulates the rate of large-scale tissue deformations during gastrulation. In the next manuscript, we find that high rates of deformation drive local eATP release, P2RY2 activation, and tissue stiffening. eATP driven tissue stiffening then slows the rate of epiboly during gastrulation. Our current manuscript provides an important foundation for the next manuscript.

We include a point-by-point response to the reviewers' comments below. We have indicated our responses in blue font.

Reviewer #1:

#1-1: While I tend to believe the data, one issue with the morpholino experiments is that the authors don't demonstrate that the P2RY2-MOs actually inhibit translation of P2RY2 mRNA. This is rather difficult without a sensitive antibody to P2RY2 protein, but the authors could show that it inhibits coinjected mRNA. They could also then show that the MO resistant P2RY2 mRNA is actually resistant. However, the rescue experiment they perform with the MO resistant P2RY2 mRNA is considered an adequate control by many colleagues. It is just that you can't tell whether it is actually rescuing the signal pathway or compensating for its absence. An alternative strategy used by many colleagues is to show the same result with a second set of P2RY2-MOs.

We appreciate the reviewer's perspective on the use of morpholinos. We have carried out our knock-down studies according to Xenopus community standards that include the use of control morpholinos and rescue (Eisen and Smith, 2008). Experimental designs involving morpholinos have been heavily debated in the field (Blum et al., 2015), but morpholinos remain an important tool in models such as zebrafish and Xenopus. Furthermore, morpholinos have been extensively used to study Xenopus epithelial biology due to the ability of these tissues to differentiate in isolation, independent of dorsal-ventral or anterior- posterior patterning processes active elsewhere (Werner and Mitchell, 2013; Brooks and Wallingford, 2015; Walentek and Quigley, 2017). In some regards, P2RY2 is a good candidate for morpholino-based knock down studies as its two homoeologs are not

expressed until after the maternal-zygotic transition (Session et al., 2016) and maternal contributions of P2RY2 are not observed in proteomics of the egg (Wühr et al., 2014).

One point of caution that is frequently raised regards activation of the innate immune system (Gentsch et al., 2018). However, our studies are carried out in the early embryo, well before there is an active immune response (Paraiso et al., 2019).

Another point of caution is the potential for off-target effects. Using BLAST, we checked for potential off-target binding sites of the morpholinos. In the *Xenopus laevis* genome, the only additional sequences had weak homology and are unlikely to be bound by the morpholinos; moreover, they are tens of kb away from any other gene. Searching the *Xenopus laevis* transcriptome also did not uncover any additional likely binding sequences in other transcripts. The only identified binding sites were the targeted translation initiation sites of p2ry2.L and p2ry2.S.

Lastly, the reviewer is correct that we do not have an antibody to confirm reduction of P2RY2, but we have a very strong phenotypic readout, e.g., loss or reduction of eATP driven contractility and rescue of contractility after injection with a MO-resistant P2RY2 mRNA. Additionally, as noted by the reviewer, injection of dominant negative G-protein signaling constructs phenocopy P2RY2 MO-injected embryos.

Regarding use of a second set of morpholinos, the reviewer is referring to the use of splice-site targeting morpholinos. This is a perfectly reasonable recommendation, however, as we state in the methods section, neither P2RY2 homeologs have introns within the coding region. (Please see the below diagram for the gene structure of both homeologs of *Xenopus laevis* p2ry2, p2ry2.L and p2ry2.S.) While both genes do have introns, in both cases the introns are confined to the 5' UTR of the mRNA (orange boxes denote the UTR); the coding sequence (blue boxes) is uninterrupted by introns. Therefore, we designed translation blocking morpholinos that bind at the indicated AUG site and interfere with translation.

We have revised the methods section to emphasize the inability to target splice acceptor sites and a statement about the limitations of morpholinos.

#1-2: The manuscript is generally well written and is easily understood, but the last sentence on page 4 needs to be written more clearly.

We thank the reviewer for pointing out this gaffe. We have revised the sentence for clarity:

"Notably, contractility occurs at the same rate after inhibition of F-actin polymerization or Rho Kinase, but is frequently eliminated when myosin II is inhibited with BBS (Fig. 2C)."

#1-3: Be consistent with the nomenclature that you use for P2Y receptors. You write about P2Y8 and P2Y11 but P2RY2. I believe that the latter is the currently accepted way of naming these receptors so it should also be P2RY8 and P2RY11

We thank the reviewer for this. We have revised the nomenclature throughout the manuscript.

Reviewer #2:

#2-1 Please reference the appropriate Niethammer papers.

We thank the reviewer for reference to these interesting papers. We have also found a 2023 paper from Bo Wang's group (Fan et al, 2023; PMID: 37480850). We have modified the introduction and added the following paragraph to the discussion to relate the findings of the current manuscript to these papers.

"Studies have identified long range signaling pathways for sensing and activating the innate immune response in the Zebrafish larval tail and regeneration in planarians. In wounded zebrafish larva, osmotic shock at the wound site (Enyedi et al., 2013), ATP release (Gault et al., 2014), production of reactive oxygen species such as hydrogen peroxide (H₂O₂) (Niethammer et al., 2009), and lipid peroxidation and production of arachidonate metabolites (Enyedi et al., 2016; Katikaneni et al., 2020) to guide leukocyte infiltration to wound sites.

Injury in the planarian also drives rapid activation of extracellular regulated kinase (Erk) across the entire body of the flatworm. Propagation of a wave of Erk activation reaches approximately 1 mm per hour. This rate is similar to the 7 mm per hour propagation of contractility in response to microfluidically streamed eATP seen previously by our group (Kim et al., 2014) albeit over a shorter range of 100 μm. Contractile responses to eATP or lysate may reflect a response to either lytic wounding and sublytic cell stress. Our previous work indicates that a range of sublytic stimuli, including laser, electrical, and lysate all produce temporally similar contractile responses (Joshi et al., 2010). However, the pathways activated downstream of osmotic shock in the Zebrafish tail wound including H₂O₂ and AA release trigger the innate immune response and recruit leukocytes to the wound area but do not appear to trigger contraction at the wound site. We note that the innate immune system in frogs is not yet developed at the embryonic stage used in our studies."

#2-2: This reviewer disagrees with the authors main conclusion, that eATP binding to P2YR mediates normal morphogenesis. Based in the data provided and literature, including a paper the authors need to cite, this pathway is more likely to mediate a wound healing response (see below). Testing whether P2YR knockdown affects normal development or/and wound healing would strengthen the paper by giving it a biological result, even if both experiments came back negative. It seems likely that in the wild, frog embryos would often suffer wounding by predators and therefore a wound response pathway would increase fitness. I suggest normal development and wound healing be given equal weight in the intro and discussion. Better, the authors should test both experimentally.

We have revised both the intro and discussion to include the potential role of P2RY2 in both wound detection and development. As mentioned above, we have a second manuscript describing the role that eATP and P2RY2 in regulating the rate of epithelial morphogenesis.

We do not know the frequency of sub-critical predation in the wild but any break in the epithelia could be catastrophic due to the prevalence of infectious agents and parasites in the aqueous environment. Wound closure would need to be rapid and efficient in stages where embryos do not have an active immune response.

#2-3: The authors should read and cite the work of Philip Niethammer's group on contractile response to eATP in embryonic epithelia in zebrafish (PMID: 25533845 and possibly also PMID: 32521230). The first paper at least is directly relevant to this submission. There is no direct overlap, rather this paper may have identified the receptor that was implied but not identified in Niethammer's work. Citing that work appropriately would force the authors to consider the hypothesis that they are working on a pathway that mediate embryo wound responses rather than assume it mediates normal development.

Please see our response to point #2-1.

#2-4: The paper would be more interesting if the authors tested whether P2Y2 plays a role in normal development and/or response to mechanical wounding. They have a knockdown method, so it should not be too hard to score effects on embryogenesis. Effects of a knockdown

on wound healing might be a bit harder to measure, but others have scored this for different molecular perturbations. Presumably they could deliver a somewhat standardized wound with a needle or laser, then quantify recovery kinetics or survival with P2YR expression intact or knocked down.

We too are very curious about the role of eATP and P2RY2 as mechanotransducers. The experiments suggested by the reviewer are feasible but still beyond the scope of the current manuscript.

References cited

- Blum, M., De Robertis, E. M., Wallingford, J. B. and Niehrs, C. (2015) 'Morpholinos: Antisense and Sensibility', *Dev Cell* 35(2): 145-9.
- Brooks, Eric R and Wallingford, John B (2015) In vivo investigation of cilia structure and function using *Xenopus* *Methods in cell biology*, vol. 127: Elsevier.
- Eisen, Judith S and Smith, James C (2008) 'Controlling morpholino experiments: don't stop making antisense'.
- Enyedi, B., Jelcic, M. and Niethammer, P. (2016) 'The Cell Nucleus Serves as a Mechanotransducer of Tissue Damage-Induced Inflammation', *Cell* 165(5): 1160-1170.
- Enyedi, B., Kala, S., Nikolich-Zugich, T. and Niethammer, P. (2013) 'Tissue damage detection by osmotic surveillance', *Nat Cell Biol* 15(9): 1123-30.
- Gault, W. J., Enyedi, B. and Niethammer, P. (2014) 'Osmotic surveillance mediates rapid wound closure through nucleotide release', *Journal of Cell Biology* 207(6): 767-782.
- Gentsch, George E, Spruce, Thomas, Monteiro, Rita S, Owens, Nick DL, Martin, Stephen R and Smith, James C (2018) 'Innate immune response and off-target mis-splicing are common morpholino-induced side effects in *Xenopus*', *Developmental cell* 44(5): 597-610. e10.
- Joshi, S. D., von Dassow, M. and Davidson, L. A. (2010) 'Experimental control of excitable embryonic tissues: three stimuli induce rapid epithelial contraction', *Exp Cell Res* 316(1): 103-14.
- Katikaneni, A., Jelcic, M., Gerlach, G. F., Ma, Y., Overholtzer, M. and Niethammer, P. (2020) 'Lipid peroxidation regulates long-range wound detection through 5-lipoxygenase in zebrafish', *Nat Cell Biol* 22(9): 1049-1055.
- Kim, Y., Hazar, M., Vijayraghavan, D. S., Song, J., Jackson, T. R., Joshi, S. D., Messner, W. C., Davidson, L. A. and LeDuc, P. R. (2014) 'Mechanochemical actuators of embryonic epithelial contractility', *Proc Natl Acad Sci U S A* 111(40): 14366-71.
- Niethammer, P., Grabher, C., Look, A. T. and Mitchison, T. J. (2009) 'A tissue-scale gradient of hydrogen peroxide mediates rapid wound detection in zebrafish', *Nature* 459(7249): 996-9.
- Paraiso, Kitt D, Blitz, Ira L, Zhou, Jeff J and Cho, Ken WY (2019) 'Morpholinos do not elicit an innate immune response during early *Xenopus* embryogenesis', *Developmental cell* 49(4): 643-650. e3.
- Session, A. M., Uno, Y., Kwon, T., Chapman, J. A., Toyoda, A., Takahashi, S., Fukui, A., Hikosaka, A., Suzuki, A., Kondo, M. et al. (2016) 'Genome evolution in the allotetraploid frog *Xenopus laevis*', *Nature* 538(7625): 336-343.
- Walentek, Peter and Quigley, Ian K (2017) 'What we can learn from a tadpole about ciliopathies and airway diseases: using systems biology in *Xenopus* to study cilia and mucociliary epithelia', *Genesis* 55(1-2): e23001.

Werner, Michael E and Mitchell, Brian J (2013) Using *Xenopus* skin to study cilia development and function *Methods in enzymology*, vol. 525: Elsevier.

Wühr, Martin, Freeman, Robert M, Presler, Marc, Horb, Marko E, Peshkin, Leonid, Gygi, Steven P and Kirschner, Marc W (2014) 'Deep proteomics of the *Xenopus laevis* egg using an mRNA-derived reference database', *Current Biology* 24(13): 1467-1475.

Third decision letter

MS ID#: jcs.263877R2

MS Title: Supracellular contractility in *Xenopus* embryo epithelia regulated by extracellular ATP and the purinergic receptor P2Y2

Authors: Sagar D Joshi; Timothy R Jackson; Lin Zhang; Carsten Stuckenholz; Lance A Davidson

Article Type: Short Report

Dear Dr Davidson,

I am happy to tell you that your manuscript has been accepted for publication in *Journal of Cell Science*, pending standard publication integrity checks.